# TIME-EXACT MULTI-BLOCKCHAIN ARCHITECTURES FOR TRUSTWORTHY MULTI-AGENT SYSTEMS

**Ivan Dorokhov, Janne Ruponen, Roman Shutsky and Andrey Nechesov** [*]
Artificial Intelligence Research Center of Novosibirsk State University, Novosibirsk, Russia
{ioandorokhov,ruponez,shutsky.roman,nechesoff}@gmail.com;

## ABSTRACT

A hierarchical multi-blockchain architecture for time-exact multi-agent system (MAS) ensure predictable transaction processing and verifiable smart contract execution. By leveraging the polynomial hierarchy and polynomial programming methodology, proposed framework integrates a reinforcement learning-based dual-mode data sharing mechanism tailored for embodied AI swarms that dynamically adapts communication fidelity from lightweight textual updates to high-fidelity sensory data sharing based on real-time context and resource constraints. Inspired by the Chinese social credit system, a reputation-based social credit mechanism is introduced to allow continuous assessment and reinforcement for agent reliability, enhancing trust and resilience within decentralized AI swarms. By combining dynamic stakeholder routing, temporal synchronization across global, regional, and local blockchain layers, and formal verification techniques, this approach addresses key limitations of conventional blockchains — including scalability, inter-agent coordination, and timing uncertainties — paving the way for next-generation, trustworthy MAS in complex domains such as urban management, supply chain logistics, and emergency response.

## 1 CHALLENGES AND APPLICATIONS OF MULTI-AGENT BLOCKCHAIN SYSTEMS

While the rise of powerful large language models (LLMs) has revolutionized approaches to versatile problem-solving, their adaption to multi-agent planning and coordination remains one of the key challenges for achieving the necessary reasoning capabilities for multi-agent systems (MAS) Chen et al. (2024). Traditional approaches to multi-agent coordination often encounter limitations in scalability and adaptability, particularly in complex and dynamic environments. A deployment of MAS increases complexity of workflows Guo et al. (2024) , where hierarchical task division becomes convoluted with type-specific agents in distributed network. Consequently, more transactions are required to facilitate collaboration among agents. Additionally, increased memory storage demands could create a bottleneck. Han et al. (2024). The use of MAS on the urban context also must comply with managerial expectations, mainly related to resource management and time allocations, where disruptions can propagate to all temporally connected processes. Further technical challenges arise in decentralized governance model, where transactional costs and performance often are unable to achieve efficiency of centralized alternatives. Time-exact multi-blockchain could provide solution in distributed system by implementing verification processes based on polynomial hierarchy. When the processing cycle is time-exact, synchronization between MAS and blockchains can be consistent and focus on agile approaches to operate within fixed time frames. Time-exact transactions, primary found in robotics and automation, financial trading, industrial control systems, traffic management and supply chain, reflect the importance of precision, real-time coordination, and synchronization.

Additionally, prospects of agents that can autonomously transact assets in service-based economy can establish further possibilities to optimize those application fields. However, as agents gain

[*]This work was supported by a grant for research centers, provided by the Analytical Center for the Government of the Russian Federation in accordance with the subsidy agreement (agreement identifier 000000D730324P540002) and an agreement with the Novosibirsk State University dated 27 December 2023 No. 70-2023-001318.

ownership of assets, issues arise from temporal ownerships. Agents receiving transactions must be able to interpret the meaning for receivables and display consistent behavior, while agents sending should embed additional notations to transactions to provide key instructions for receiving agents. Papi et al. (2022)

Maintenance and operations of distributed and secure MAS becomes computationally intensive as more agents are employed, especially when transactions are bind to strict time constraints. Considering this, integration of MAS within existing blockchains would have to compromise temporal coordination in effective hierarchical task division. Here, highly deterministic temporal coordination requires guarantees in precise execution times for allocated transactions across distributed agents without introducing excessive computational overhead. In addition, consensus mechanisms in existing blockchain systems introduce variable latencies that can disrupt time-sensitive multi-agent operations, making it difficult to establish reliable service-level agreements (SLAs). The lack of axiomatization in the first-order logic previously has limited the deeper understanding about fundamental axioms of the blockchain theory to build time-exact applications Goncharov & Nechesov (2023b). Consequently, this has led to the introduction of blockchain trilemma, highlighting that blockchains can satisfy only two conditions among scalability, decentralization and security Conti et al. (2019). Blockchain implementations have prioritized decentralization and security conditions over scalability Gangwal et al. (2023). As the scalability has been compromised, latencies in transaction times and high-volume transactions have failed to achieve performance of traditional systems - payment systems being perhaps the most concrete example of this Croman et al. (2016).

Reinforcement learning (RL) has emerged as a promising technique, enabling agents to learn effective strategies through interaction. However, optimizing task allocation and resource management in large-scale MAS remains an active area of research. Fang et al. propose a genetic algorithm-enhanced proximal policy optimization (GAPPO) algorithm to address these challenges Fang et al. (2025). GAPPO combines deep reinforcement learning with genetic algorithms, focusing on both efficient task allocation and energy optimization. Their results demonstrate GAPPO's advantages compared to traditional RL algorithms, achieving reductions in task completion time while maintaining energy efficiency across agent fleets. GAPPO underscores the benefits of hybrid approaches that integrate RL with other optimization techniques to create robust and efficient multi-agent coordination frameworks. Future research directions identified by Fang et al. include further optimization of energy management strategies, validation in real-world applications, and integration of advanced optimization techniques to enhance adaptability Fang et al. (2025). This highlights the ongoing need for research into effective and scalable multi-agent coordination frameworks, particularly those that address resource constraints and dynamic environments.

This paper presents a novel approach to designing time-exact and trustworthy MAS by leveraging a hierarchical multi-blockchain architecture. The research focused on the detailed design of algorithmic control for MAS hierarchical and time-exact blockchain. Specifically, the key research objectives and contributions included a) development of RL-based dual-mode data sharing mechanism in time-exact multi-blockchain for enabling embodied AI agents to dynamically switch between lightweight textual updates and high-fidelity sensory data sharing, and B) integration of a social credit mechanism within trusted and resilient decentralized AI swarms into the RL reward function to promote trustworthy agent behavior and prevention of malicious activities.

## 2 HIERARCHICAL MULTI-BLOCKCHAIN ARCHITECTURE AND INTEGRATION WITH P-COMPLETE PROGRAMMING REALIZATION

The proposed solution leverages formal reasoning capabilities described in Nechesov (2024) to enable individual agents to provide explanations for their actions, contributing to the overall trustworthiness of the system. The concept of polynomial-time complexity is crucial for the development of time-exact multi-blockchain architectures for trustworthy MAS Goncharov & Nechesov (2023a). In such systems, predictable and bounded execution times are essential for both individual agent computations and inter-agent interactions across multiple blockchains. Polynomial complexity provides a mathematical guarantee of efficiency, ensuring that computations remain tractable even as system complexity increases. This predictability is vital for achieving time-exactness, as it allows developers to reason about the system's behavior and establish timing guarantees. Furthermore, polynomial complexity contributes to overall system trustworthiness by ensuring that agents and smart contracts

execute reliably and within predictable time frames (for example, civic participation in city governance Nechesov & Ruponen (2024)). These underlying principles were discussed in Goncharov et al. (2024), main focus being on polynomial complexity Nechesov & Goncharov (2024) within single programs or computations. Same principles can be applied to the design of efficient agents and smart contracts within a multi-blockchain environment.

## 2.1 POLYNOMIAL HIERARCHY AND MULTI-AGENTS

The Polynomial Hierarchy (PH) is a classification of decision problems based on their computational complexity, extending the concept of NP and co-NP problems by incorporating alternating quantifiers. When applied to MAS, this model offers a structured approach to categorizing the computational complexity of tasks that agents must perform, both individually and during interactions with other agents within the system. The polynomial hierarchy can help in defining the complexity of agent behaviors, ensuring that agents can operate efficiently even as the number of agents and interactions grows. In a multi-agent environment, agents can be designed to operate at different levels of the polynomial hierarchy, which reflects the complexity of their decision-making capabilities and the types of tasks they can address. At the lower levels, agents perform computations that involve straightforward decision processes, such as simple optimization tasks, which can be solved in polynomial time (P-level problems). This also involves the activation of agents based on their idle state to ensure higher utilization rate of available agents. These agents are responsible for relatively basic, predictable behaviors, like collecting data, monitoring environmental conditions, and taking action based on direct inputs.

In the context of MAS, the polynomial hierarchy provides a structured framework to categorize agent tasks by complexity. For instance, basic data collection and direct sensor processing tasks fall within P-level problems, ensuring fast, predictable execution. More complex tasks, such as multi-agent negotiation or coordinated decision-making, can be modeled at higher levels (NP-level), where although finding a solution may be challenging, verifying proposed solutions remains efficient. This classification aids in designing MAS protocols that guarantee bounded execution times and predictable inter-agent interactions.

The Polynomial Hierarchy (PH), a cornerstone of classical computational complexity, has traditionally been studied in the context of single-machine computation. However, its application to distributed computing, particularly within the local model, offers new insights and advantages. Reiter has extended the classical PH to a distributed setting, where networked computers communicate via synchronous message-passing to collectively solve a decision problem about their network topology Reiter (2024). Their *local view* of the PH imposes constraints on both the number of communication rounds (constant) and the local computation performed by each computer (polynomial in input and message size). This distributed PH, defined through alternating quantifiers assigned by players, generalizes the classical PH and complexity classes P and NP. Intriguingly, key results from classical complexity theory, such as the Cook-Levin theorem and Fagin's theorem, can be extended to this distributed setting Reiter (2024). Furthermore, separating complexity classes, a notoriously difficult problem in the classical case, becomes more tractable in the distributed context. Reiter demonstrate the infinitude of their distributed PH, a result that remains open for the classical PH Reiter (2024). This work highlights the potential of the PH as a tool for analyzing the complexity of distributed problems and introduces quantifier alternation as a novel approach to measuring locality in distributed computing.

In many real-world MAS applications, actions must occur within strict time windows, and deviations can lead to significant consequences, from missed deadlines to system failures. One of the key targets is to avoid situations where race conditions would occur, as these can lead to unpredictable delays or even system failures, thus compromising the timeliness and reliability of time-critical operations within the MAS. Time-exact processing ensures that transactions, which represent interactions or operations within the MAS, are executed precisely when they are scheduled, respecting temporal constraints and dependencies. Moreover, integrating P-complete programming languages Goncharov & Nechesov (2022) within a time-exact MAS offers significant advantages in terms of predictability and verifiability. P-complete problems, by definition, are solvable in polynomial time, making them inherently well-suited for time-critical applications. In the context of a MAS, this means that agents can efficiently compute solutions to complex problems within bounded timeframes. These solutions can be verified in polynomial time, providing a robust mechanism to ensure

the correctness of actions and transactions. Combination of the expressive power of P-complete programming with the time-exactness of the underlying architecture opens up possibilities for integrating formal verification techniques, allowing for even stronger guarantees about the correctness and timeliness of agent interactions. Using P-complete programming, agents can engage in complex decision-making processes while still providing verifiable evidence of their actions.

## 2.2    Time-Exact Smart Contract Execution and Coordination

The implementation of time-exact multi-blockchain approach for MAS should focus on formalizing a smart contract structure to provide base instructions compatible with preferred mechanisms for consensus, scheduling, agent coordination, and interchain operations. Smart contracts govern the interactions within the multi-blockchain system. The solution involves the key functions, data structures, and logic used to manage time-critical transactions, agent interactions, and data sharing, with a particular focus on how time-exactness is enforced at the smart contract level. This includes defining how temporal constraints are represented within the contracts and how the contracts interact with the underlying scheduling mechanisms.

Each smart contract encapsulates the logic for a specific type of agent task or interaction. Contracts are classified based on their computational complexity, ensuring all contracts remain within P or NP-complete complexity. This ensures that verification remains feasible within polynomial time constraints. *P-level contracts* handle tasks solvable in polynomial time. Examples include simple data collection, basic sensor processing, or direct responses to inputs. The execution time of these contracts can be precisely bounded. These contracts are analogous to agents performing simple optimization or rule-based actions. *NP-level contracts* address tasks whose solutions can be verified in polynomial time. These contracts might implement verification logic for proposed solutions to more complex problems, such as negotiation outcomes or coordination strategies. While finding the solution might be difficult, verifying its correctness is efficient. These contracts are relevant to agents evaluating potential plans or strategies. For NP-level contracts, the use of P-complete verification logic can ensure that any proposed solution can be verified in polynomial time. The system is restricted to tasks that remain within NP-complete complexity, preventing the need for verification beyond polynomial constraints. Problems requiring verification beyond P are excluded to maintain time-bounded execution feasibility.

A conceptual smart contract in this context involves temporal variables for allocation periods, agent functions for submitting exchange requests, a verification function (P-computable) for validation, and logic to interact with the transaction scheduler. If an agent submits a conflicting exchange request that overlaps with an already allocated transactions, the smart contract employs a priority resolution mechanism. This mechanism may rely on predefined criteria such as agent reputation, bid value, or a first-come, first-served approach, depending on the logic of the contract. In cases where multiple agents compete for the same resource, the contract initiates a resolution process, either through an automated arbitration function or a time-limited negotiation round where agents can adjust their bids or requests. To prevent deadlocks, the contract enforces strict time-bounded execution by integrating with the transaction scheduler. If a resolution is not reached within the allocated time window, the contract either defaults to a fallback allocation strategy (e.g., random selection or weighted distribution) or resets the process, allowing agents to resubmit modified requests. This ensures that execution remains predictable, avoiding indefinite contention and maintaining system responsiveness.

Optimal consensus protocol for different blockchain levels provides rules for validation contract state changes. A consensus protocol is responsible for ensuring agreement among agents on the state of smart contracts and the validity of transactions. Different consensus mechanisms should be employed at different blockchain levels to optimize for the specific requirements of each layer (local, regional, global). At the local level, where temporary blockchains are formed for rapid consensus on time-critical events, speed and efficiency are prioritized. Candidates for local consensus protocols include Practical Byzantine Fault Tolerance (PBFT) or Raft, as these are known for their relatively low latency and ability to handle a smaller number of participants Castro et al. (1999); Ongaro & Ousterhout (2014). The specific choice will depend on the trade-off between fault tolerance and speed required for the particular application. The regional level requires a balance between speed and scalability, as it manages interactions among a larger number of agents within a specific geographic area or functional domain. Potential candidates for regional consensus include variations

of PBFT or other protocols designed for higher throughput while maintaining reasonable latency. The global level focuses on long-term consistency and security, managing the overall state of the system and aggregating information from the regional blockchains. Here, protocols that prioritize security and decentralization should be considered, even if they might have slightly higher latency, such as Proof-of-Stake (PoW), Proof-of-Stake (PoS) or variations thereof.

The integration of the consensus protocol with the smart contracts is essential for ensuring that contract execution is both valid and consistent across the network. When a smart contract function is invoked, the consensus protocol ensures that all participating agents execute the contract logic in a deterministic manner. The resulting state changes are then recorded on the appropriate blockchain (local, regional, or global), with the consensus mechanism guaranteeing consistency. Specifically, for time-critical contracts, the consensus process must complete within the time bounds defined in the contract itself. For NP-level contracts, the consensus ensures that submitted solutions are verifiable in polynomial time. The protocol integrates with the verification function in the smart contract, guaranteeing that only valid solutions are accepted. The consensus mechanism must be aware of the time constraints associated with smart contracts. Achieving time-exactness in a distributed consensus environment presents several challenges. Network delays can impact the speed of message propagation and consensus achievement. The architecture mitigates this by using localized blockchains for time-critical operations, reducing the number of participants involved in consensus. Furthermore, the choice of consensus protocol at each level is influenced by its tolerance to network latency. Multiple agents might attempt to execute the same contract concurrently. The consensus mechanism must handle this concurrency correctly, ensuring that transactions are processed in a consistent order and that race conditions are avoided. This may involve techniques like locking or optimistic concurrency control.

A scheduling protocol defines how transactions are scheduled and prioritized to ensure time-exact execution. This protocol focuses on prioritizing and executing transactions within strict temporal constraints. Corresponding approach considers transaction dependencies, resource allocation, and dynamic system changes to establish time-exact transaction scheduling mechanism. The scheduler should be able to inform the consensus protocol about time-critical transactions, and the consensus protocol should be able to prioritize these transactions accordingly. A combination of data structures and algorithms are used to manage and schedule transactions effectively. A priority queue can be used to store pending transactions, ordered by their deadlines or priorities. Transactions with earlier deadlines or higher importance are placed at the front of the queue. This allows the scheduler to quickly identify and process the most time-critical transactions. A directed graph represents dependencies between transactions. An edge from transaction A to transaction B indicates that B cannot be executed until A has completed. This graph allows the scheduler to identify and respect transaction dependencies, ensuring that transactions are executed in the correct order. A scheduling algorithm that combines priority-based scheduling with dependency analysis is utilized. The scheduler selects the highest-priority transaction from the queue, checks if all its dependencies have been satisfied, and if so, schedules it for execution. If dependencies are not yet met, the transaction is kept in the queue until its dependencies become available. Variations of Earliest Deadline First (EDF) or Least Laxity First (LLF) scheduling algorithms, adapted for a distributed environment, are considered.

The scheduler is designed to adapt to dynamic changes in the system, such as the arrival of new transactions, changes in agent states, or unexpected events. If a new, highly time-critical transaction arrives, the scheduler can preempt a currently executing lower-priority transaction and re-schedule it to a later time. This allows the system to respond to urgent events and ensure that the most critical transactions are always executed promptly. The dependency graph can be updated dynamically as new transactions arrive or existing transactions are modified. The scheduler can then re-evaluate the transaction schedule based on the updated dependencies. If a node responsible for executing a transaction fails, the scheduler can re-assign the transaction to another node, ensuring that it is still executed within its deadline.

Interactions between blockchains is handled by an interchain protocol, providing optimized message routing between the different blockchains in a hierarchical multi-blockchain architecture. For time-critical information, the protocol must incorporate mechanisms to prioritize and expedite message delivery with a dedicated queuing. The protocol should also include mechanisms for handling message loss or delays along with automatic retries. Addressing latency and maintaining consistency across blockchains are key design considerations. For most communication, an asynchronous approach should be used. This allows blockchains to continue processing transactions without waiting

for immediate responses, improving overall throughput. For specific time-critical operations synchronous communication can be used with load balancer. Consistency across blockchains can be retained by including message ordering, timestamping, and verification. Security is a crucial aspect of the interchain communication protocol. Authentication mechanisms are defined my authorization rules in control access to specific resources or functionalities. Further, security benefits from encryption and digital signatures of interchain messages. Smart contracts can initiate interchain communication to request services or share information with other smart contracts on different blockchains. The scheduler works with the communication protocol to ensure that time-critical messages are delivered and processed in a timely manner. For certain interchain operations that require consensus, the communication protocol interacts with the consensus mechanism to ensure agreement among the relevant blockchains.

Once smart contract details and involved protocols are formalized, given rules enable time-exact communications and synchronizations for agent coordination. The agent coordination framework orchestrates the interactions of agents within a time-exact multi-blockchain system. This framework integrates the underlying mechanisms (smart contracts, consensus, scheduling, and interchain communication) to enable agents to effectively coordinate their actions while respecting temporal constraints. Key components of this framework include the RL-based dual-mode data sharing mechanism, the social credit system, and the algorithms for dynamic stakeholder routing and temporary blockchain formation.

Time constraints are explicitly integrated into the agent coordination process. Time-aware coordination protocols consider time constraints when scheduling agent actions and message exchanges. Agents are aware of deadlines associated with their tasks and coordinate their actions to ensure that deadlines are met. The agent coordination framework works closely with the transaction scheduler to ensure that agent actions and inter-agent communication respect the overall system's time constraints.

## 3 TRUST, RESILIENCE, AND ASYNCHRONOUS GLOBAL INTEGRATION IN DECENTRALIZED MULTI-BLOCKCHAIN SYSTEMS

In this section, advanced mechanisms are consolidated for ensuring robust trust, scalable coordination, and time-exact data sharing within MAS. The framework combines dynamic local consensus through temporary blockchain formations, social credit–based trust mechanisms, and reinforcement learning (RL)–driven dual-mode data sharing. These elements are integrated atop a hierarchical multi-blockchain architecture, where local, regional, and global layers interact via asynchronous updates. The following subsections detail each component of this approach.

### 3.1 TRUST AND RESILIENCE IN SOCIAL SWARM-DECENTRALIZED AI SYSTEMS

**Ensuring Trust and Preventing Falsifications:** To ensure trust in the system, a comprehensive approach is employed to combine cross-verification mechanisms, a social rating system, and the immutability provided by blockchain. When an event (for example, a fallen tree) is detected, sensor data is transmitted into the regional network, where several independent agents—including nearby vehicles and specialized cleanup services—participate in verifying the event. If an agent transmits high-fidelity data that meets expected parameters and this information is corroborated by other participants, its social rating is increased. Conversely, if inconsistencies or attempts to inject falsified information (such as deepfakes) are detected, the agent is penalized by a reduction in its rating. This socially oriented rating mechanism not only dynamically assesses each participant's reliability but also prevents the spread of false data, as a low rating restricts an agent's participation in critical processes. Moreover, recording all transactions and events on the blockchain ensures transparency and provides an immutable audit trail, further reinforcing trust in the system.

**Social Swarm-Decentralized AI: Mechanisms of Resilience and Control:**

The 'social swarm' concept leverages a decentralized social credit system inspired by Chinese social credit models to dynamically assess and update each agent's trustworthiness. In our framework, every agent's social credit score is continuously refined through peer verification—quantitatively measured via metrics such as trust convergence rate and credit score stability (see Section 3.4 and Table 1). This distributed mechanism enables dynamic role allocation, where agents with higher

reliability are given precedence in critical operations, thereby ensuring high system resilience and mitigating risks of collusion or misinformation.

A key feature of the proposed system is the concept of social swarm-decentralized artificial intelligence. In this approach, a multitude of autonomous agents—each assigned a social rating (inspired by Chinese social credit systems applied to AI swarms)—work together in a decentralized manner. Each agent's rating is based on the quality of the information it disseminates and the results of cross-verification with other network participants. This distributed structure reduces the risk of power concentration and ensures high system resilience: if an agent behaves maliciously or transmits falsified data, its social rating rapidly declines, limiting its role in critical processes. The "divide and conquer" strategy is implemented through dynamic role allocation and continuous inter-agent verification, making collective subversion nearly impossible. Should any agent attempt to disrupt the system, other agents—relying on their verification algorithms and social rating mechanism—will promptly isolate it. This approach not only enhances the reliability of information exchange but also fosters a robust decentralized environment where decisions are made collectively by all participants. As a result, the system exhibits high efficiency and stability even under highly dynamic conditions and in the face of potential external threats.

## 3.2 Algorithmic Control of Dual-Mode Data Sharing in Embodied AI Networks

In urban environments populated by diverse embodied AI—ranging from autonomous vehicles and drones to humanoid androids—precise coordination is paramount. A time-exact multi-blockchain framework is extended with a context-aware data sharing strategy that combines dual-mode communication with reinforcement learning (RL)–based dynamic mode switching. This integration not only creates a decentralized "blockchain memory" for persistent environmental records but also optimizes network resource usage by ensuring that only the necessary level of detail is transmitted in a given context.

Each agent continuously monitors its surroundings via onboard sensors. Transmitting all raw sensor data would overwhelm the network; therefore, this approach supports two complementary communication modes:

1. **Lightweight Updates:** Agents broadcast concise, textual summaries (e.g., "Tree blocking road at (x, y, z)") over the global and regional blockchain layers, creating a low-overhead, persistent record of environmental conditions.

2. **High-Fidelity Sensory Sharing:** In critical scenarios—such as collision avoidance or emergency maneuvers—agents switch to transmitting rich sensory data (e.g., high-resolution images, LIDAR scans, audio recordings) to enable immediate response.

Furthermore, the decision to switch between lightweight and high-fidelity data sharing is directly influenced by the agent's social credit score. As detailed in Section 3.5, agents with higher trust levels are prioritized for high-fidelity transmission, ensuring that critical data is disseminated only by reliable entities and thereby enhancing overall system integrity.

### 3.2.1 Dynamic Mode Switching via Reinforcement Learning

A central challenge is determining, in real time, which communication mode to use based on local context. This is addressed by equipping each agent with an RL module that evaluates its state and selects the optimal mode. The agent's state at time $t$ is represented as:

$$s_t = \{p(t),\, T(t),\, E(t),\, L(t)\},$$

where:

- $p(t)$ is the agent's position and proximity to others,
- $T(t)$ denotes the current operational task,
- $E(t)$ is an event indicator (e.g., detection of hazards),
- $L(t)$ represents the current network and computational load.

Based on $s_t$, the agent selects an action from:

$$a_t \in \{\texttt{Lightweight Sharing}, \texttt{High-Fidelity Sharing}\}.$$

The RL policy $\pi(a|s)$ (with exploration, e.g., $\epsilon$-greedy) is trained to maximize the expected cumulative reward:

$$J(\pi) = \mathbb{E}\left[\sum_{t=0}^{\infty} \gamma_{RL}^t \, r(s_t, a_t)\right],$$

with the reward function defined as:

$$r(s_t, a_t) = \alpha \, \text{Safety}(s_t, a_t) + \beta \, \text{Efficiency}(s_t, a_t) - \gamma \, \text{ResourceCost}(a_t).$$

Here, $\gamma$ (in the reward function) is a weighting parameter for cost, while $\gamma_{RL}$ is the RL discount factor.

**Pseudocode:** The following pseudocode outlines the integrated RL-enhanced dual-mode data sharing process, including dynamic stakeholder routing and fidelity function generation for high-fidelity actions.

---

**Algorithm 1** Integrated RL-Enhanced Dual-Mode Data Sharing with Dynamic Fidelity Generation

---

1: **Input:** Set of agents $A$, initial RL policy $\pi$, reward parameters $\alpha, \beta, \gamma$, RL discount factor $\gamma_{RL}$, learning rate $\eta$
2: **for** each time step $t$ **do**
3:     **for** each agent $i \in A$ **do**
4:         **State Acquisition:**

$$s_i(t) \leftarrow \{p_i(t),\, T_i(t),\, E_i(t),\, L_i(t)\}$$

5:         **Action Selection:** $a_i(t) \sim \pi(a|s_i(t))$                    ▷ Use $\epsilon$-greedy exploration
6:         **if** $a_i(t) = \texttt{High-Fidelity Sharing}$ **then**
7:             Initiate detailed sensory sharing.
8:             **Stakeholder Routing:**

$$H_i = \mathcal{R}(E_i(t), p_i(t), r)$$

9:             **for** each stakeholder $h \in H_i$ **do**
10:                 Generate dynamic fidelity function:

$$f_h = \mathcal{F}(E_i(t), p_i(t), h)$$

11:             **end for**
12:         **else**
13:             Broadcast lightweight textual update.
14:         **end if**
15:         **Reward Computation:**

$$r_i(t) = \alpha \, \text{Safety}(s_i(t), a_i(t)) + \beta \, \text{Efficiency}(s_i(t), a_i(t)) - \gamma \, \text{ResourceCost}(a_i(t))$$

16:         **Policy Update:** Update Q-value using:

$$Q(s_i(t), a_i(t)) \leftarrow Q(s_i(t), a_i(t)) + \eta \left[ r_i(t) + \gamma_{RL} \max_a Q(s_i(t+1), a) - Q(s_i(t), a_i(t)) \right]$$

▷ Standard Q-learning update; update $\pi$ accordingly

17:     **end for**
18: **end for**

---

### 3.2.2 INTEGRATION WITH HIERARCHICAL BLOCKCHAIN ARCHITECTURE

The RL-controlled dual-mode sharing is embedded within the hierarchical multi-blockchain framework to guarantee both scalability and time-exact data dissemination:

- **Global Layer:** Routine, lightweight updates are posted here, building a long-term ledger of environmental conditions and trends.

- **Regional Layer:** Dynamically formed blockchains (using clustering techniques such as DBSCAN Ester et al. (1996)) ensure that agents in high-density areas exchange context-relevant updates without rigid geographic boundaries.
- **Local Layer:** In safety-critical scenarios, agents employ high-fidelity sharing over local channels (blockchain-based or otherwise) to enable immediate coordination.

### 3.2.3 DYNAMIC STAKEHOLDER-DRIVEN EVENT RESOLUTION

For instance, when an autonomous vehicle detects a fallen tree:

1. **Sensory Data Interpretation:** An onboard LLM processes raw sensor data $D$ to generate an event descriptor:
$$E = \mathcal{M}(D),$$
where $E$ includes the event type, location $\ell$, and road identifier $r$.

2. **Stakeholder Routing:** The agent queries the regional blockchain:
$$H = \mathcal{R}(E, \ell, r) = \{h \in A : \|p_h - \ell\| \leq \delta \wedge h.\text{road} = r \wedge M(h)\},$$
selecting only relevant stakeholders.

3. **Dynamic Fidelity Generation:** The onboard LLM (using RAG Lewis et al. (2020)) produces a tailored fidelity function:
$$f_h = \mathcal{F}(E, \ell, h),$$
so that nearby vehicles receive high-fidelity data while distant agents receive succinct updates.

4. **Temporary Event Blockchain Formation:** A localized blockchain network is instantiated among these stakeholders for rapid, high-fidelity data exchange and consensus on event resolution. Once resolved, the event record is archived for future reference and training Brown et al. (2020).

**Example:** Consider an autonomous vehicle approaching an urban core along a forest road. Initially, it transmits lightweight updates to the global ledger. Upon detecting a fallen tree, the RL module—assessing the heightened risk and increased local density—triggers high-fidelity sharing. The vehicle's LLM processes the event, routes relevant stakeholders via the regional blockchain, and dynamically sets fidelity functions so that nearby vehicles and service providers receive detailed sensory data. A temporary blockchain is then formed for real-time coordination until the obstacle is cleared.

### 3.2.4 SUMMARY

By combining dual-mode data sharing with an RL-based dynamic mode switching module, the framework ensures that embodied AI agents communicate contextually and efficiently within a hierarchical multi-blockchain environment. The unified RL mechanism governs decisions across both routine and emergency scenarios, while integration with blockchain layers guarantees time-exact, scalable, and verifiable data dissemination. This cohesive design underpins a decentralized, distributed intelligence that meets the rigorous demands of urban MAS. Before deploying such system into the real world, time-accelerated virtual city simulation can serve as a testing ground Nechesov et al. (2025)

### 3.3 SYNCHRONOUS LOCAL CONSENSUS VIA DYNAMIC TEMPORARY BLOCKCHAINS

Traditional synchronous updates across an entire MAS can overload the network and induce latency. Instead, the system partitions agents into localized clusters that form temporary blockchains on demand to ensure rapid and reliable local decision-making when a critical event occurs. This is achieved through the following steps:

- **Dynamic Clustering:** Agents identify relevant stakeholders using a routing function $\mathcal{R}(\cdot)$ based on context (e.g., proximity, event type, network load).

- **Temporary Blockchain Formation:** The selected agents form a temporary, localized blockchain. Within this network, they execute a synchronous consensus protocol (e.g., PBFT or Raft) that ensures all participating agents share a consistent state and decision. This is particularly important for decisions regarding high-fidelity data sharing.

- **Decoupled Global Integration:** The locally reached consensus (and any policy or state updates) is then integrated asynchronously into the global blockchain. This decoupling allows local decisions to be rapid and robust while preventing global network congestion.

## 3.4 SOCIAL SWARM-DECENTRALIZED AI: TRUST AND RESILIENCE MECHANISMS

Ensuring data integrity and trust is critical in a decentralized environment. The framework implements a social credit system inspired by Chinese social credit models, which evaluates the reliability of all transmitted updates—whether high-fidelity sensory data or low-fidelity textual updates.

- **Peer Verification:** Each agent $i$ is associated with a set of neighboring agents $N_i$ that assess the quality and contextual appropriateness of its transmitted updates. Let $v_{ij}(t)$ be an indicator function defined as:

$$v_{ij}(t) = \begin{cases} 1, & \text{if agent } j \text{ confirms that agent } i \text{ transmitted accurate and context-appropriate data,} \\ 0, & \text{otherwise.} \end{cases}$$

  This verification process applies to both high-fidelity data and low-fidelity textual updates. If an agent unnecessarily spams low-fidelity updates into the global or regional network, or uses low fidelity when high fidelity is required, these misbehaviors will be penalized through a lower verification score.
  The aggregated verification score is computed as:

$$V_i(t) = \frac{1}{|N_i|} \sum_{j \in N_i} v_{ij}(t).$$

- **Social Credit Update:** An agent's social credit score $S_i(t)$ is updated using an exponential moving average:

$$S_i(t+1) = S_i(t) + \eta_S \Big( V_i(t) - S_i(t) \Big),$$

  where $\eta_S$ is the learning rate. This mechanism rewards agents that transmit accurate and context-appropriate updates, while penalizing those that frequently send inappropriate or spammy updates, regardless of whether the update is high- or low-fidelity.

- **Integration with RL:** The updated social credit $S_i(t)$ is incorporated into the RL reward function. Consequently, an agent's decision-making is directly influenced by its reliability:

$$r'(s_i, a_i) = \alpha \, \text{Safety}(s_i, a_i) + \beta \, \text{Efficiency}(s_i, a_i) - \gamma \, \text{ResourceCost}(a_i) + \delta \, S_i(t),$$

  where $\alpha$, $\beta$, $\gamma$, and $\delta$ are weighting parameters that balance operational safety, efficiency, resource cost, and trust. This formulation ensures that any misbehavior (e.g., spamming low-fidelity updates when inappropriate) adversely affects the agent's overall reward.

## 3.5 INTEGRATED RL-BASED DUAL-MODE DATA SHARING WITH SOCIAL CREDIT

This approach leverages an RL module to dynamically choose between two data-sharing modes:

- **Lightweight Sharing:** Under normal conditions, agents broadcast concise textual updates to conserve bandwidth.

- **High-Fidelity Sharing:** In high-risk or emergency scenarios, agents switch to transmitting rich sensory data (e.g., high-resolution images, LIDAR scans) and trigger the formation of a temporary blockchain for synchronous consensus.

Each agent's state is defined as:

$$s_i(t) = \{p_i(t), \, T_i(t), \, E_i(t), \, L_i(t), \, S_i(t)\},$$

where:

- $p_i(t)$ is the spatial position,
- $T_i(t)$ denotes the current task,
- $E_i(t)$ is an event indicator,
- $L_i(t)$ represents local network and computational load,
- $S_i(t)$ is the agent's social credit.

The RL policy $\pi(a|s)$ selects an action $a_i(t)$ from the set $\{\texttt{Lightweight Sharing}, \texttt{High-Fidelity Sharing}\}$. The modified reward function is:

$$r'(s_i, a_i) = \alpha \,\text{Safety}(s_i, a_i) + \beta \,\text{Efficiency}(s_i, a_i) - \gamma \,\text{ResourceCost}(a_i) + \delta \, S_i(t),$$

where $\alpha$, $\beta$, $\gamma$, and $\delta$ are weighting parameters that balance operational safety, efficiency, resource cost, and trust.

In the system, each agent decides whether to share high-fidelity data or only lightweight updates based on its local context. This decision is governed by a reinforcement learning (RL) module. In addition, to promote trustworthiness in the decentralized AI swarm, a social credit mechanism is designed to adjusts each agent's reward based on peer verification. This subsection outlines the integrated approach.

### 3.5.1 AGENT STATE AND ACTION SPACE

At each time step $t$, an agent $i$ observes a state vector:

$$s_i(t) = \{p_i(t), \, T_i(t), \, E_i(t), \, L_i(t), \, S_i(t)\},$$

where:

- $p_i(t)$ is the agent's position and proximity to others,
- $T_i(t)$ represents its current operational task,
- $E_i(t)$ is an event indicator (e.g., detection of hazards),
- $L_i(t)$ denotes the current network and computational load,
- $S_i(t)$ is the agent's current social credit score.

Based on $s_i(t)$, the agent selects an action:

$$a_i(t) \in \{\texttt{Lightweight Sharing}, \texttt{High-Fidelity Sharing}\},$$

using an RL policy $\pi(a|s)$ (with, for example, $\epsilon$-greedy exploration).

### 3.5.2 SOCIAL CREDIT UPDATE MECHANISM

When an agent shares high-fidelity data, its output is verified by a set of neighboring agents $N_i$. For each event, define the peer verification score as:

$$V_i(t) = \frac{1}{|N_i|} \sum_{j \in N_i} v_{ij}(t),$$

where $v_{ij}(t) \in \{0, 1\}$ is an indicator function (1 if agent $j$ confirms the accuracy of $i$'s data, 0 otherwise). The social credit score is then updated using an exponential moving average:

$$S_i(t+1) = S_i(t) + \eta_S \Big( V_i(t) - S_i(t) \Big),$$

with $\eta_S \in (0, 1)$ as the social credit learning rate.

### 3.5.3 MODIFIED RL REWARD FUNCTION

The base RL reward function for an agent's decision is defined as:

$$r(s_t, a_t) = \alpha \,\text{Safety}(s_t, a_t) + \beta \,\text{Efficiency}(s_t, a_t) - \gamma \,\text{ResourceCost}(a_t).$$

This reward is extended to include the social credit score:

$$r'(s_t, a_t) = \alpha \,\text{Safety}(s_t, a_t) + \beta \,\text{Efficiency}(s_t, a_t) - \gamma \,\text{ResourceCost}(a_t) + \delta \, S_i(t),$$

where $\delta$ is a weighting parameter that controls the influence of the social credit on the overall reward.

### 3.5.4 INTEGRATED RL UPDATE AND PSEUDO-CODE

The RL policy is updated using the modified reward $r'$ in a standard Q-learning framework:

$$Q(s_i(t), a_i(t)) \leftarrow Q(s_i(t), a_i(t)) + \eta \left[ r'_i(t) + \gamma_{RL} \max_{a'} Q(s_i(t+1), a') - Q(s_i(t), a_i(t)) \right],$$

where $\eta$ is the learning rate and $\gamma_{RL}$ is the RL discount factor.

The following pseudo-code integrates the dual-mode sharing decision, dynamic stakeholder routing for high-fidelity data, and social credit update, encapsulating the entire process from state acquisition and action selection through local consensus and social credit updates to achieve asynchronous global integration.

---

**Algorithm 2** Integrated RL-Enhanced Dual-Mode Data Sharing with Social Credit

---

1: **Input:** Set of agents $A$, initial RL policy $\pi$, reward parameters $\alpha$, $\beta$, $\gamma$, $\delta$, RL discount factor $\gamma_{RL}$, learning rates $\nu$, $\nu_S$
2: **for** each time step $t$ **do**
3:     **for** each agent $i \in A$: **do**
4:         $s_i(t) \leftarrow \{p_i(t), T_i(t), E_i(t), L_i(t), S_i(t)\}$             $\triangleright$ State Acquisition
5:         $a_i(t) \sim \pi(a|s_i(t))$         $\triangleright$ Action Selection (e. g., via $\epsilon$-greedy exploration)
6:         **if** $a_i(t) = \,'\text{High} - \text{FidelitySharing}'$ **then**    $\triangleright$ Initiate high-fidelity data transmission.
7:             $H_i = R(E_i(t), p_i(t), r)$             $\triangleright$ Dynamic Stakeholder Routing
8:             **for** each stakeholder $h \in H_i$ **do**
9:                 $f_h = F(E_i(t), p_i(t), h)$        $\triangleright$ Generate dynamic fidelity function
10:                 Form Temporary Blockchain: Agents in $H_i$ reach synchronous consensus.
11:             **end for**
12:         **else**
13:             Broadcast lightweight textual update.
14:         **end if**
15:         $V_i(t) = \frac{1}{|N_i|} \sum_{j \in N_i} v_{ij}(t)$             $\triangleright$ Social Credit Update
16:         $S_i(t+1) = S_i(t) + \nu_S(V_i(t) - S_i(t))$
17:         $r'_i(t) = \alpha \cdot \text{Safety}(s_i(t), a_i(t)) + \beta \cdot \text{Efficiency}(s_i(t), a_i(t)) - \gamma \cdot \text{ResourceCost}(a_i(t)) + \delta \cdot S_i(t)$             $\triangleright$ Reward Computation
18:         $Q(s_i(t), a_i(t)) \leftarrow Q(s_i(t), a_i(t)) + \nu \left[ r'_i(t) + \gamma_{RL} \cdot \max_{a'} Q(s_i(t+1), a') - Q(s_i(t), a_i(t)) \right]$   $\triangleright$ RL Update
19:         Local consensus outcomes and policy updates are transmitted
20:         asynchronously to the global blockchain.                $\triangleright$
21:     **end for**
22: **end for**

---

### 3.6 BENEFITS AND IMPACT ON REAL-WORLD APPLICATIONS

By combining synchronous local consensus, a trust-enhancing social credit system, and RL-based dual-mode data sharing, the framework offers several key benefits. Firstly, it provides an approach for improved scalability and reduced latency as localized temporary blockchains enable fast and synchronized decisions among agents, while avoiding network load with asynchronous global interactions. Secondly, the social credit mechanism provides incentives for accurate and context-sensitive data-sharing, where trust and resilience are not compromised. Thirdly, RL-driven mode switching allows agents to dynamically adjust communication fidelity based on real-time environmental and operational contexts. Consequently, impact on real-world applications should particularly benefit complex, time-sensitive scenarios such as urban management, emergency response, and supply chain logistics.

These benefits are achieved through an integrated approach combining multiple mechanisms. A dual-mode sharing system, governed by a reinforcement learning (RL) module, dynamically selects between high-fidelity and lightweight data based on local context. A social credit system ensures trust by incorporating peer verification, which updates each agent's credit score and influences its RL reward. Finally, RL integration modifies the reward function to guide Q-learning updates, incentivizing both efficient information sharing and reliable agent behavior.

By aligning individual agent incentives with overall system integrity, this framework promotes a trustworthy, resilient, and efficient decentralized AI swarm. In summary, this unified section integrates applicable concepts into a comprehensive approach that ensures both the precision and reliability required for next-generation MAS. This cohesive design not only addresses the limitations of traditional blockchains but also paves the way for scalable, trustworthy, and context-aware decentralized networks.

# 4 EVALUATION SYSTEM

The evaluation of proposed system focuses on theoretical analysis and scenario-based assessment, as specified in research objectives. Comprehensive set of performance vectors and evaluation criteria are introduced to assess the system's feasibility, potential performance, and theoretical bounds.

## 4.1 PERFORMANCE QUADRANT

A performance quadrant system is proposed as a structured approach to categorizing key performance dynamics in distributed and adaptive systems by evaluating them across two critical dimensions: *temporal efficiency* and *system adaptability*. The X-axis, representing temporal efficiency, distinguishes between vectors that have an immediate impact on system behavior and those whose influence accumulates over time. Vectors positioned on the left side of the quadrant exhibit short-term impact, meaning their effects are observable almost instantaneously, often in real-time operations. Conversely, vectors on the right side are associated with long-term impact, emerging as a result of system learning, adaptation, or large-scale structural evolution. The Y-axis measures system adaptability, differentiating between vectors that require dynamic, real-time responsiveness (high adaptability) and those that reflect more static, infrastructure-dependent behaviors (low adaptability). High-adaptability vectors typically involve machine learning, real-time consensus mechanisms, or system optimizations, while low-adaptability vectors are tied to foundational properties that evolve slowly over time. Fig. 1 provides visual presentation to demonstrate quadrants.

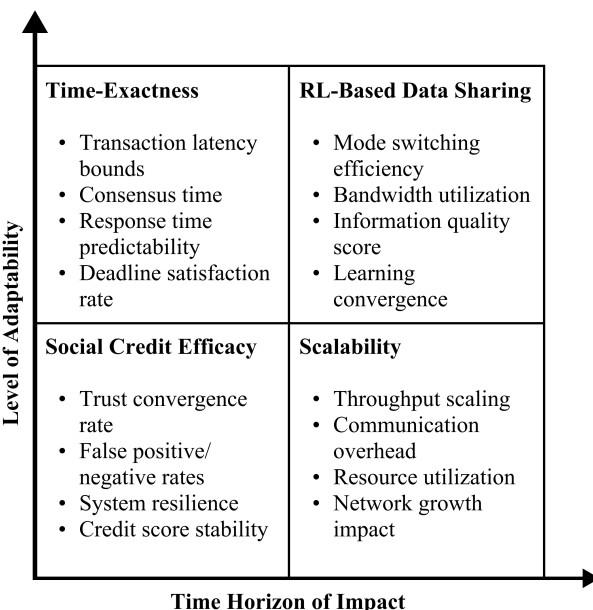

Figure 1: Performance Vectors Quadrant.

By distinguishing between short-term and long-term impact while considering the degree of adaptability required, this classification helps in understanding the trade-offs between real-time responsiveness, learning-based optimizations, trust stability, and scalability in adaptive and distributed

environments. It serves as a foundation for evaluating system efficiency and designing architectures that balance immediate operational needs with sustainable long-term growth.

Each quadrant of the performance evaluation framework corresponds to distinct aspects of the proposed architecture. Specifically, the Time-Exactness quadrant reflects the efficiency of smart contract scheduling and consensus protocols across the local, regional, and global blockchain layers. The RL-Based Data Sharing quadrant quantifies the adaptability of the dual-mode communication mechanism driven by reinforcement learning. Social Credit Efficacy directly measures the reliability of the peer verification and social credit system, while Scalability assesses the system's capacity to maintain performance as the number of agents increases. These connections are further detailed in Table 1, which defines specific measurement methodologies, success criteria, and comparative baselines, thereby ensuring that the theoretical performance metrics are directly linked to operational outcomes. However, it should be noted that vectors introduced in this paper were limited to narrow scope - in real-time environment it is anticipated that the number of vectors will be significantly higher.

**Time-Exactness Vectors** consists of highly adaptable vectors with immediate impact. These indicators are crucial for systems that require fast response times, predictable latency, and rapid consensus mechanisms. The system must be capable of adjusting in real time to meet stringent performance constraints, making adaptability a key factor in ensuring stability. For example, response time predictability ensures that operations remain within expected latency bounds, preventing performance degradation. Similarly, consensus time measures how quickly agents or nodes reach agreement, a critical factor in distributed networks where delays can cause inconsistencies. These vectors are particularly relevant in applications such as blockchain validation, real-time bidding systems, and mission-critical control systems, where fast reaction times are paramount.

To assess time-critical aspects, transaction latency bounds are measured by timestamping transaction lifecycle events, using network monitoring and blockchain logs. Consensus time is tracked by timing the distributed agreement process on transactions, monitoring message exchanges between nodes. Response time predictability involves analyzing the statistical distribution of response times collected through automated scripts. Finally, deadline satisfaction rate is determined by comparing completion times with deadlines, using system logs for accurate tracking.

**RL-Based Data Sharing Vectors** represents vectors that rely on RL or other adaptive strategies to optimize performance over time. While these vectors still demand high adaptability, their impact is realized gradually as the system learns and improves efficiency. For instance, learning convergence ensures that models refine their decision-making over multiple iterations, leading to more optimal bandwidth utilization and intelligent mode-switching strategies. Unlike time-exactness vectors, which require instant precision, these indicators benefit from iterative improvements and long-term adjustments. Such performance considerations are crucial in AI-driven networks, autonomous decision-making platforms, and self-optimizing communication protocols.

Evaluating the efficiency of adaptive data sharing involves several metrics. Mode switching efficiency is assessed by tracking the speed, accuracy, and overhead of switching between data sharing modes. Bandwidth utilization is monitored using network tools to capture packet transmission rates. Information quality score uses a predefined metric to evaluate the relevancy and accuracy of shared data. Learning convergence tracks the reinforcement learning model's performance over time, analyzing metrics like reward accumulation and policy stability.

**Social Credit Efficacy Vectors** collects vectors that prioritize short-term assessment of system trustworthiness but operate within rigid evaluation frameworks. These indicators measure how efficiently a system detects and assesses trustworthy behavior but do not significantly adapt over time. Trust convergence rate and credit score stability illustrate this principle, as they ensure that users or entities are rapidly assigned a trustworthiness score based on predefined rules rather than dynamic learning. While a system may quickly evaluate behavior, the underlying scoring logic remains relatively fixed. This quadrant is particularly relevant for applications such as fraud detection, reputation management, and risk assessment, where immediate identification of unreliable actors is necessary, but the mechanisms governing evaluation evolve slowly.

To measure the reliability of the social credit system, the analysis tracks how quickly stable trust scores are assigned, using logs to analyze trust convergence patterns. False positive/negative rates are determined by comparing system assessments with immutable base data. System resilience is

evaluated through attack simulations, measuring the impact on the social credit scores. Credit score stability is assessed by analyzing the variance of credit scores over time, using logged updates.

**Scalability Vectors** encompasses vectors that measure large-scale system performance and resource management over extended periods. These indicators exhibit low adaptability, as they depend on deep architectural decisions rather than real-time modifications. Scalability concerns, such as throughput scaling and communication overhead, become increasingly significant as the system expands, yet their optimization typically requires structural changes rather than immediate adjustments. The impact of these vectors emerges gradually as the network grows, making them essential for evaluating long-term system sustainability. Applications such as decentralized networks, cloud infrastructure scaling, and distributed databases rely on these vectors to ensure that expanding workloads can be handled efficiently without compromising overall performance.

Scalability is evaluated by measuring throughput scaling, observing how transaction processing rates change with increasing users or nodes through load testing. Communication overhead is monitored by capturing network traffic data and CPU utilization. Resource utilization tracks the consumption of CPU, memory, and storage using system monitoring tools. Network growth impact is assessed through simulations, observing the effect of network expansion on other scalability metrics.

## 4.2 EVALUATION METHODS

The evaluation approach consists of four key considerations for quadrant vectors. *Measurement Methodology* describes how the metric is captured in a system while *Theoretical Bounds* sets expected performance limits in best- and worst-case scenarios. *Comparative Baseline* sets reference points from existing systems or models, and *Success Criteria* declares thresholds that indicate acceptable performance. The comparison between quadrants vectors with success criteria, measurement methodologies, and comparative baselines is displayed in table 1.

| Quadrant Vectors | Measurement Methodology | Success Criteria | Comparative Baseline |
|---|---|---|---|
| *Time-Exactness* | - Response time variance
- Consensus time
- Deadline satisfaction rate | - Response time deviation within 1% of expectations
- 99.9%+ deadlines met | Traditional real-time systems (e.g., blockchain validation) |
| *RL-Based Data Sharing* | - Mode switching efficiency
- Bandwidth utilization
- Learning convergence rate | - 95%+ model convergence within predefined iterations
- 5%+ efficiency improvement per iteration | Static allocation-based non-RL systems |
| *Social Credit Efficacy* | - Trust convergence rate
- False positive/negative rates
- Credit score stability | - Trust convergence within $2\sigma$
- False positive/negative rates below 2% | Traditional reputation management systems |
| *Scalability* | - Throughput scaling
- Communication overhead
- Resource utilization | - Linear or better throughput scaling (O(n))
- <10% degradation per doubling network size | Existing large-scale distributed networks |

Table 1: Comparison of Vectors for System Performance Evaluation

The *Measurement Methodology* for each quadrant focuses on the specific vectors that need to be captured in order to evaluate system performance effectively. For time-exactness vectors, response time variance, consensus time, and deadline satisfaction rates are key indicators to monitor real-time performance, ensuring systems meet stringent timing requirements. In the case of RL-based data sharing, the primary vectors include mode switching efficiency, bandwidth utilization, and learning convergence rates, all of which assess the adaptability and efficiency of the system during its learning process. Social credit efficacy requires monitoring trust convergence rates, false positive/negative rates, and credit score stability to ensure that the system remains reliable while minimizing any unfair penalties or rewards. Scalability vectors focus on throughput scaling, communication overhead, and resource utilization to understand how the system adapts to larger loads without significant degradation in performance.

The *Success Criteria* are closely related to Service Level Agreements (SLA), which serve as contractual guarantees between consumers and providers in service-oriented systems, ensuring reliability, availability, and performance Bianco et al. (2008). For time-exactness vectors, a 1% response time

deviation and 99.9%+ deadline adherence align with industry standards for real-time systems, where even minor timing inconsistencies can cause cascading failures. For instance, Amazon S3 guarantees 99.9% monthly uptime as part of its SLA commitments Bianco et al. (2008). RL-based data sharing, which focuses on long-term adaptability, follows established reinforcement learning patterns by targeting 5%+ efficiency improvement per iteration and 95%+ model convergence accuracy within predefined epochs. Social credit efficacy, being a short-term but low-adaptability system, requires trust convergence within $2\sigma$ and a false positive rate below 2%, ensuring stability while minimizing unfair penalties. Scalability, a long-term challenge with low adaptability, necessitates at least linear throughput scaling (O(n)) and performance degradation under 10% when the network doubles, maintaining efficiency under growing demands.

*Comparative Baselines* provide reference points for performance comparisons with existing systems or models. Time-exactness vectors are compared to traditional real-time systems, such as blockchain validation processes, where high precision in timing is critical. RL-based data sharing systems are evaluated against static allocation-based non-RL systems to benchmark the performance improvements provided by the adaptive nature of reinforcement learning. Social credit efficacy is compared to traditional reputation management systems, which also aim to assess trustworthiness, but often without the dynamic adaptability inherent in social credit systems. Finally, scalability vector are compared against existing large-scale distributed networks, which provide insight into how the system will perform as it scales across more nodes or users. These baselines ensure that system performance can be accurately gauged and improved over time, providing valuable context for each metric.

Further, Queuing models and Markov Decision Processes (MDPs) are powerful tools which provide formal support for practical and approximate decision-making in adaptive and distributed systems Boucherie & Van Dijk (2017). These models are mapped across different quadrants based on two dimensions: temporal efficiency (short-term vs. long-term impact) and system adaptability (high vs. low adaptability). Analyzing these models through system performance vectors helps balance immediate responsiveness with long-term strategic goals. In the time-exact quadrant, queuing models are ideal for real-time resource allocation in systems that require quick adjustments, such as service counters in a busy environment. MDPs can also be applied here for dynamic decision-making, adjusting strategies based on evolving data, like in adaptive traffic control or call center routing. In the RL-based data sharing quadrant, queuing models can predict evolving system needs, such as bandwidth requirements in adaptive networks. However, MDPs are more fitting as they provide an iterative framework for long-term decision optimization, making them useful in AI-driven networks and self-optimizing systems where strategies improve over time. In the social credit quadrant, queuing models help allocate resources efficiently in predictable environments, such as trust or fraud detection systems. MDPs are less applicable here, as they are more suited to systems requiring ongoing learning, whereas this quadrant focuses on rigid, short-term evaluations. In the scalability quadrant, queuing models are critical for managing system growth and resource allocation over time, as seen in cloud infrastructure scaling. MDPs are less relevant in this context, though they could be used to explore long-term strategies in large systems with evolving needs.

## 5 DISCUSSION

While the proposed hierarchical multi-blockchain framework introduces innovative mechanisms for time-exact, verifiable, and scalable multi-agent systems (MAS) coordination, several limitations must be acknowledged. First, managing multiple blockchain layers—especially when forming dynamic temporary blockchains for localized consensus—introduces non-negligible computational and communication overhead. As the number of agents increases, this layered architecture may encounter latency challenges, particularly during peak operations or in highly dynamic environments. Moreover, integrating reinforcement learning for dynamic mode switching, although promising, may require extensive tuning to cope with real-world uncertainties, such as noisy sensor inputs or unforeseen event dynamics. Finally, while the reputation-based social credit mechanism enhances trust, its reliance on peer verification might be susceptible to collusion or biased evaluations within heterogeneous agent networks.

To mitigate these challenges, future research should explore several avenues. Advanced reinforcement learning techniques —including deep multi-agent RL Foerster et al. (2016) and meta-learning

approaches Hospedales et al. (2021) — could further improve adaptive decision-making under uncertainty. Refining temporary blockchain consensus protocols (for instance, by blending PBFT with faster, lightweight algorithms) may help reduce latency and enhance scalability. Additionally, incorporating edge-computing strategies can offload processing from central nodes, enabling faster local consensus and data handling. Enhancements to the social credit mechanism, such as incorporating anomaly detection or distributed fairness algorithms, can further reduce the risk of manipulation by adversarial agents.

The proposed framework effectively bridges rigorous theoretical constructs—including the polynomial hierarchy and P-complete verification—with practical implementations in decentralized MAS. This synthesis demonstrates that time-exact transaction processing is achievable even in complex, dynamic environments. By aligning local consensus, dynamic data sharing, and trust management within a multi-blockchain context, the framework opens new research directions in both distributed AI and blockchain-enabled architectures.

For real-world deployment, several practical issues must be considered. Network reliability, hardware constraints, and regulatory requirements are critical factors that will influence implementation. Designers must balance the trade-offs between high-fidelity sensory data sharing and lightweight textual updates to prevent network congestion while meeting strict timing requirements. Moreover, pilot studies in simulated environments—such as virtual urban settings or controlled industrial zones—will be essential to validate the theoretical performance vectors and ensure system adaptability and resilience under diverse operational conditions.

# 6 CONCLUSIONS

This paper has introduced a novel hierarchical multi-blockchain architecture for time-exact MAS that integrates a reinforcement learning-based dual-mode data sharing mechanism with a reputation-driven social credit system. By leveraging principles from the polynomial hierarchy and formal verification techniques, the framework ensures predictable transaction processing, dynamic stakeholder routing, and precise temporal synchronization across local, regional, and global blockchain layers. In doing so, it addresses key challenges in scalability, inter-agent coordination, and timing uncertainties, paving the way for more trustworthy and resilient deployments in complex domains.

Moreover, the evaluation framework provides a comprehensive approach to assessing the performance of distributed and adaptive systems across multiple dimensions. By categorizing key vectors into four distinct quadrants —Time-Exactness, RL-Based Data Sharing, Social Credit Efficacy, and Scalability— it becomes possible to identify the trade-offs between short-term operational needs and long-term system adaptability. The proposed measurement methodologies, success criteria, and comparative baselines offer a structured way to benchmark performance, ensuring that the system meets both real-time responsiveness and sustainable growth requirements. Furthermore, the integration of queuing models and MDPs enhances the ability to optimize decision-making, balancing immediate performance with long-term system evolution. This framework serves as a robust foundation for future research and system design, guiding the development of adaptive, scalable systems that can efficiently manage complex, dynamic environments. Overall, by merging formal complexity theory with practical multi-agent coordination strategies, the framework paves the way for next-generation decentralized systems with significant potential for applications such as urban management, emergency response, and supply chain logistics.

Future research should concentrate on several key areas. First, investigating more advanced reinforcement learning algorithms —such as multi-agent deep RL and adaptive meta-learning— could further enhance the system's responsiveness in dynamic environments. Second, refining consensus protocols and exploring alternative blockchain architectures may reduce latency and improve throughput in large-scale deployments. Third, extensive real-world testing, including large-scale simulations and pilot implementations, is essential to validate performance vectors and fine-tune system parameters. Finally, further theoretical work on extending the distributed polynomial hierarchy model may provide deeper insights into the complexity and efficiency of decentralized decision-making in MAS.

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
