# OpenReview forum: "Time-Exact Multi-Blockchain Architectures for Trustworthy Multi-Agent Systems"
_mathai.club/MathAI/2025/Conference — MathAI 2025 Oral_

### Official Review · Reviewer_DHXt · 2025-02-26
**TIME-EXACT MULTI-BLOCKCHAIN ARCHITECTURES FOR TRUSTWORTHY MULTI-AGENT SYSTEMS. Reviewer recommends to include it in the Program of the International conference “Mathematics of Artificial Intelligence” 24-28 March 2025, Sochi with its publication.**

**Rating:** 9
**Confidence:** 4

**Review:**

The article is devoted to the actual problem of choosing and justifying such hierarchical multi-blockchain architecture for time-exact multi-agent system (MAS), which ensures predictable transaction processing and verifiable smart contract execution. The anonymous authors propose to use the polynomial hierarchy and polynomial programming methodology to integrate a reinforcement learning based dual-mode data sharing mechanism tailored for embodied AI swarms. It dynamically adapts communication fidelity from lightweight textual updates to high-fidelity sensory data sharing based on real-time context and resource constraints. A reputation-based social credit mechanism is introduced to allow continuous assessment and reinforcement for agent reliability, enhancing trust and resilience within decentralized AI swarms (this has some analogies with the known Chinese social credit system). The proposed approach combines dynamic stakeholder routing, temporal synchronization across global, regional, and local blockchain layers, and formal verification techniques to overcome key limitations of conventional blockchains — including scalability, inter-agent coordination, and timing uncertainties. It paves the way for next-generation, trustworthy MAS in complex domains such as urban management, supply chain logistics, and emergency response.
        The article contains a wide list of references (26 appropriate literary sources) which allows you to become familiar with many additional details of the approach presented.
       The reviewer agrees with the following final provisions of the article. Future research should concentrate on several key areas. First, investigating more advanced reinforcement learning algorithms —such as multi-agent deep RL and adaptive meta-learning— could further enhance the system’s responsiveness in dynamic environments. Second, refining consensus protocols and exploring alternative blockchain architectures may reduce latency and improve throughput in large-scale deployments. Third, extensive real-world testing, including large-scalesimulations and pilot implementations, is essential to validate performance vectors and fine-tunesystem parameters. Finally, further theoretical work on extending the distributed polynomial hierarchy model may provide deeper insights into the complexity and efficiency of decentralized decision-making in MAS.
       The presented theme is not simple, of course, but the article is written quite clearly and in a clear language. The quality, clarity, originality and significance of this work are high. I like this article and I recommend to include it in the Program of the International conference “Mathematics of Artificial Intelligence” 24-28 March 2025, Sochi with its publication.

---

### Official Review · Reviewer_MNSh · 2025-02-27
**A well-structured paper presenting a novel multi-blockchain framework for time-exact MAS coordination; recommended for acceptance despite minor scalability and trust challenges.**

**Rating:** 8
**Confidence:** 3

**Review:**

This paper presents a novel hierarchical multi-blockchain framework designed to ensure time-exact transaction processing and verifiable smart contract execution in multi-agent systems (MAS). By integrating polynomial hierarchy theory and polynomial programming methodologies, the authors propose a reinforcement learning-based dual-mode data-sharing mechanism tailored for embodied AI swarms. The approach dynamically adjusts communication fidelity based on real-time context and resource constraints. Additionally, a reputation-based social credit mechanism inspired by the Chinese social credit system is introduced to enhance trust and resilience within decentralized AI swarms.


## Strengths of the Paper:

- Innovative Approach: the proposed hierarchical multi-blockchain framework effectively addresses key challenges related to scalability, inter-agent coordination, and timing uncertainties. The integration of polynomial complexity theory with blockchain-based MAS coordination is a significant contribution.

- Adaptive Data Sharing Mechanism: the reinforcement learning-based dual-mode data-sharing approach optimizes communication efficiency by dynamically selecting between lightweight textual updates and high-fidelity sensory data, improving resource allocation.

- Trust and Resilience Enhancements: the inclusion of a reputation-based social credit mechanism provides a novel method for continuous assessment and reinforcement of agent reliability, contributing to system robustness.


## Areas for Improvement:

- Computational and Communication Overhead: the multi-layered blockchain structure introduces additional processing and communication demands. While temporary local blockchains aim to mitigate latency, the performance of such a system under high agent densities remains a concern.

- Potential Risks of Social Credit Mechanism: while the reputation-based mechanism aims to enhance trust, its reliance on peer verification may be vulnerable to collusion or biased evaluations. Addressing these risks with anomaly detection or fairness-enhancing techniques could strengthen the proposed approach.

## Recommendation:

This paper presents a well-researched, theoretically grounded, and practically relevant approach to integrating blockchain architectures with MAS coordination. The combination of formal verification, adaptive data sharing, and trust management mechanisms represents a significant step forward in decentralized AI systems. While certain aspects require further investigation, such as computational efficiency, fairness in reputation mechanisms, and large-scale scalability, these do not undermine the overall merit of the work.

Based on the originality, technical soundness, clarity, and potential impact of the proposed framework, I recommend acceptance of this paper for inclusion in the program of “Mathematics of Artificial Intelligence”.

---

### Official Review · Reviewer_212d · 2025-02-27
**Time-Exact Multi-Blockchain Architectures for Trustworthy Multi-Agent Systems**

**Rating:** 10
**Confidence:** 4

**Review:**

In the paper convincing combination of the dynamic stakeholder routing, temporal synchronization across blockchain layers, and formal verification techniques was developed that overcomes the main limitations of blockchain - scalability, inter-agent coordination, and timing uncertainties. A hierarchical multi-blockchain architecture for time-exact multi-agent system was used for predictable transaction processing and verifiable smart contract execution. Based on the original author rezults on polynomial hierarchy and polynomial programming, the framework was proposed that integrates a reinforcement learning, based on dual-mode data sharing mechanism tailored for AI swarms, that dynamically adapts communication fidelity from lightweight textual updates to high-fidelity sensor data sharing, based on real-time context and resource constraints. Chinese social credit system was also provided to allow continuous assessment for agent reliability, enhancing trust and resilience within decentralized AI swarms.

Strengths
1. Originality and novelty, based on the original results on polynomial computability.
2. Original combination of very different approaches.
3. Combination of the latest developments in AI - blockchains, AI swarms, polynomial hierarchy and polynomial computability.

---

### Official Review · Reviewer_wM4L · 2025-02-27
**Time-Exact Multi-Blockchain Architectures for Trustworthy Multi-Agent Systems**

**Rating:** 9
**Confidence:** 3

**Review:**

Paper presents a novel hierarchical multi-blockchain architecture designed to address key challenges in multi-agent systems (MAS): scalability, inter-agent coordination, timing uncertainties. The proposed framework integrates reinforcement learning (RL)-based dual-mode data sharing, a reputation-based social credit mechanism, and formal verification techniques to ensure predictable transaction processing and verifiable smart contract execution.

Strengths:

1. Adaptive Communication Strategy. The RL-based dual-mode data sharing mechanism is a significant advancement, allowing agents to adapt their communication fidelity based on real-time needs, which is crucial for resource-constrained environments.

2. Trust and Resilience. The social credit mechanism enhances trust and resilience in decentralized AI swarms, addressing the challenge of malicious behavior and data falsification.

3. Formal Guarantees. The use of polynomial-time complexity and P-complete programming provides strong theoretical foundations for time-exactness and verifiability, ensuring that the system can handle complex tasks within predictable time frames.

Weaknesses:

1. Computational Overhead. The proposed architecture, particularly the dynamic formation of temporary blockchains, may introduce significant computational and communication overhead, especially as the number of agents increases. This could lead to latency issues in highly dynamic environments.

Conclusion:

The paper presents a highly innovative and theoretically sound framework for time-exact multi-agent systems using hierarchical multi-blockchain architectures. Overall, the framework has significant potential for applications in complex, time-sensitive domains, but further research is needed to address scalability, computational overhead, and potential vulnerabilities in the social credit system.

---

### Official Review · Reviewer_zmc2 · 2025-02-27
**TIME-EXACT MULTI-BLOCKCHAIN ARCHITECTURES FOR TRUSTWORTHY MULTI-AGENT SYSTEMS**

**Rating:** 10
**Confidence:** 3

**Review:**

The topic of the article is suitable for the Math2025 conference. The article relates to a highly sought-after area of ​​AI, where the problems of large language models, multi-blockchains, trusted AI, multi-agent AI systems, mathematical logic, programming languages ​​and applications are concentrated. Well-known implementations of blockchains are Bitcoin or Ethereum. The article is devoted to the development of the direction to multi-blockchains in various areas, from finance, healthcare and supply chain logistics to urban management with emergency response. The use of several cooperatively functioning blockchains in multi-blockchains improves data privacy, reduces transaction costs, and provides more scalable, secure and optimal solutions. The work is an important step towards axiomatization of multi-blockchain theory (blockchain theory has recently been axiomatized in first-order logic). Axiomatization will allow formulating precise, rigorous and unambiguous statements about the behavior of multi-blockchains, will provide a clear understanding of their potential advantages and limitations, will facilitate the development and deployment of multi-blockchain systems
The paper proposes a new hierarchical software framework with several blockchains for multi-agent systems, which uses reinforcement learning, a formalized fragment of social credit, principles of polynomial hierarchy (P-complete programming) and formal verification methods.
The implemented social credit system as a rating mechanism dynamically evaluates the reliability of each participant in the multi-agent system, blocks the dissemination of false data (a low rating limits the agent's participation in critical processes). Transparency (recording of all transactions in the blockchain) provides the necessary audit trail, which strengthens trust in the system. The mathematical guarantee of the effectiveness of the framework is provided by the polynomial complexity of the programming languages ​​used, including with significant scaling and an increase in the complexity of the system. The framework uses smart contracts that control interactions in the system, effectively connects strict theoretical constructions (polynomial hierarchy, P-complete verification, and others) with practical implementations in decentralized multi-agent systems. The framework proposed in the article actually implements the concept of decentralized artificial intelligence of a social swarm.
The report on this article will be very interesting for the Math2025 conference, the article can be highly appreciated and accepted into the conference program.

---

### Official Review · Reviewer_nfPg · 2025-02-27
**Review of the paper: "Time-Exact Multi-Blockchain Architectures for Trustworthy Multi-Agent Systems"**

**Rating:** 7
**Confidence:** 3

**Review:**

The paper proposes a comprehensive solution for multi-agent systems (MAS) using a hierarchical multi-blockchain architecture. The authors use modern methods such as polynomial hierarchy, P-complete programs, reinforcement learning (RL), and social credit mechanisms. However, the paper has some gaps in the logic of presentation and coherence of the text, which may make it difficult for the reader to understand:
- Section 2 introduces the concept of **"polynomial hierarchy" (PH)**, but its application in the context of multi-agent systems is not sufficiently explained. It is unclear how exactly PH helps in solving MAS problems.
- Section 3.1 introduces the concept of **"social swarm"**, but it is not further developed in the paper.
- Section 3.2 introduces **dual-mode data exchange**, but does not explain how it integrates with the social credit mechanism described in Section 3.1.
- Section 4.1 introduces the **performance quadrant**, but does not explain how exactly these vectors relate to the proposed architecture and how they will be measured in practice. However, the proposed performance quadrant allows us to evaluate the system's performance along several key parameters, which is a strength of the paper.

The work has high relevance for areas such as urban infrastructure management, logistics, and emergency response.

---

### Decision · Program_Chairs · 2025-03-08

**Decision:**

Accept (Oral)

**Comment:**

Your article has been accepted and you can give a talk on the article. All articles will be sorted by rating and within the available conference places one author from each article will be invited. If there are not enough places, then you will either have the opportunity to speak remotely or come at your own expense!